# Ketogenic Diet Protects from Experimental Colitis in a Mouse Model Regardless of Dietary Fat Source

**DOI:** 10.3390/nu16091348

**Published:** 2024-04-29

**Authors:** Lotta Luiskari, Jere Lindén, Markku Lehto, Hanne Salmenkari, Riitta Korpela

**Affiliations:** 1Department of Pharmacology, Faculty of Medicine, University of Helsinki, 00014 Helsinki, Finland; 2Human Microbiome Research Program, Faculty of Medicine, University of Helsinki, 00014 Helsinki, Finland; 3Department of Veterinary Biosciences, Faculty of Veterinary Medicine, University of Helsinki, 00014 Helsinki, Finland; jere.linden@helsinki.fi; 4Finnish Centre for Laboratory Animal Pathology, Helsinki Institute of Life Science, University of Helsinki, 00014 Helsinki, Finland; 5Folkhälsan Institute of Genetics, Folkhälsan Research Center, 00290 Helsinki, Finland; markku.lehto@helsinki.fi (M.L.); hanne.salmenkari@helsinki.fi (H.S.); 6Department of Nephrology, University of Helsinki and Helsinki University Hospital, 00290 Helsinki, Finland; 7Research Program for Clinical and Molecular Metabolism, Faculty of Medicine, University of Helsinki, 00014 Helsinki, Finland

**Keywords:** intestinal inflammation, ketogenic diet, dietary fat, colitis

## Abstract

While ketogenic diets (KDs) may have potential as adjunct treatments for gastrointestinal diseases, there is little knowledge on how the fat source of these diets impacts intestinal health. The objective of this study was to investigate how the source of dietary fat of KD influences experimental colitis. We fed nine-week-old male C57BL/6J mice (*n* = 36) with a low-fat control diet or KD high either in saturated fatty acids (SFA-KD) or polyunsaturated linoleic acid (LA-KD) for four weeks and then induced colitis with dextran sodium sulfate (DSS). To compare the diets, we analyzed macroscopic and histological changes in the colon, intestinal permeability to fluorescein isothiocyanate−dextran (FITC–dextran), and the colonic expression of tight junction proteins and inflammatory markers. While the effects were more pronounced with LA-KD, both KDs markedly alleviated DSS-induced histological lesions. LA-KD prevented inflammation-related weight loss and the shortening of the colon, as well as preserved *Il1b* and *Tnf* expression at a healthy level. Despite no significant between-group differences in permeability to FITC–dextran, LA-KD mitigated changes in tight junction protein expression. Thus, KDs may have preventive potential against intestinal inflammation, with the level of the effect being dependent on the dietary fat source.

## 1. Introduction

Inflammatory bowel diseases (IBDs), namely, Crohn’s disease and ulcerative colitis, are chronic conditions characterized by inflammation of the gastrointestinal tract. Symptoms of these diseases include abdominal pain, chronic diarrhea, rectal bleeding, and weight loss. There is no known cure for IBDs, and their prevalence is rising worldwide [1]. They decrease both life quality and expectancy of patients [2]—thus, new forms of treatment are needed.

A low-carbohydrate, high-fat ketogenic diet (KD) is a possible adjunct treatment for IBDs [3]. On KD, the body enters a state of ketosis wherein the levels of circulating ketone bodies, such as β-hydroxybutyrate (BHB), increase. Most tissues of the body can utilize these molecules for energy production. While this is particularly important for the brain in low-glucose states, the intestine also uses ketone bodies readily as an energy source [4]. Ketone bodies function not only as energy sources, but they are also anti-inflammatory [5] and decrease oxidative stress [6]. KD has shown efficacy in preclinical models of intestinal diseases. In experimental settings, the diet can protect from colon cancer [7], improve irritable bowel syndrome [8], and ameliorate dextran sodium sulfate (DSS)-induced colonic inflammation [9,10]. However, the results are not entirely uniform. In one study, KD worsened experimental colitis by modifying microbiota and its metabolites and increasing intestinal permeability [11].

In addition to the amount of dietary fat, the fat source and the fatty acid composition of the diet can also influence intestinal physiology. Some studies indicate that linoleic acid (LA), an essential omega-6 polyunsaturated fatty acid, can, in large quantities, predispose the intestinal barrier to insults, whereas saturated fatty acids (SFAs), specifically medium-chain triglycerides, may not have the same effect [12,13]. According to one hypothesis, this might be due to differences in lipid peroxidation [13]. Previously, we have observed that, in the context of KD, intestinal permeability in healthy animals remains unchanged after four weeks of feeding regardless of the fat source but KD high in SFAs upregulates colonic tight junction (TJ) proteins and intestinal alkaline phosphatase activity whereas a LA-rich KD does not [14]. However, it is unknown whether these are compensatory mechanisms or beneficial changes.

Due to the discrepancy between animal studies on KD and intestinal inflammation, we aimed to explore how KD affects DSS-induced colitis and whether the effect is dependent on the dietary fat source. The mice were fed either a low-fat control diet or one of the two KDs—high in SFAs (SFA-KD) or high in LA (LA-KD)—for four weeks before the induction of colitis. To compare the effects of the diets, we analyzed macroscopic and histological changes in the colon, colonic expression of inflammatory markers and TJ proteins, and intestinal permeability.

## 2. Materials and Methods

### 2.1. Animal Experiment

The experiment was conducted under the approval of the animal research board of the Regional State Administrative Agency for Southern Finland (ESAVI/9377/2019). Thirty-six eight-week-old male C57BL/6J mice were purchased from Scanbur (Karlslunde, Denmark). Dietary interventions were started after seven days of acclimatization. The mice were housed individually and kept under a 12 h light–dark cycle, at 20 ± 2 °C and 50–60% humidity. Access to food and water was unrestricted.

Before the start of the study, the mice were randomly allocated to four groups based on the diet and the DSS treatment: healthy on a control diet (CD) (*n* = 8), DSS on a control diet (DSS-CD) (*n* = 8), DSS on a SFA-rich KD (DSS-SFA-KD) (*n* = 10), and DSS on a LA-rich KD (DSS-LA-KD) (*n* = 10, one of which had to be sacrificed after three weeks of the intervention due to progressive weight loss). The study diets were custom-made (Envigo, Indianapolis, IN, USA) and matched for protein and micronutrients. Their macronutrient compositions are presented in Table 1, and the exact diet formulations are provided in Appendix A.

On day 28, colitis was induced in the three DSS groups by replacing regular drinking water with DSS (40 kDa, TdB Labs, Uppsala, Sweden) solution (2.5% *w/v*). After five days of DSS administration, the animals were given regular water for two days, and sacrificed on day 35. From the start of the administration, weight, stool consistency, and visibility of fecal blood were monitored daily. Based on a previously described scoring system [15], each of these parameters was given a score from 1 to 4, from which disease activity index (DAI) was calculated as an average. Food and fluid consumptions were measured daily. Figure 1 illustrates the set-up of the animal experiment.

### 2.2. Collection of Samples

On day 35, animals were killed under isoflurane (4%, Vetflurane, Virbac, Carros, France) anesthesia by exsanguination. Blood was drawn from *vena cava* into EDTA tubes (Kisker, Steinfurt, Germany). Samples were centrifuged at 2000× *g* for 15 min at 4 °C. The separated plasma was snap-frozen in liquid nitrogen and stored in −80 °C.

After confirming the death of the animal, the colon was removed, and its length (from cecocolic orifice to rectum) was measured. Both before and after opening it longitudinally, the colon was photographed for macroscopic evaluation. Intestinal contents were flushed off with ice cold 0.9% NaCl solution. Three 0.5–1-cm-long samples were collected from the distal part of the colon for histological and biochemical analyses. Samples for the latter were frozen in liquid nitrogen and stored at −80 °C.

### 2.3. Measurement of Intestinal Permeability

Four hours before euthanasia, animals were given fluorescein isothiocyanate−dextran (FITC−dextran) (4 kDa, TdB Labs) solution (600 mg/kg, 125 mg/mL) via a gastric gavage. For the analysis, plasma was diluted in PBS-T (136 mM NaCl, 8 mM Na_2_HPO_4_, 2.7 mM KCl, 4.46 mM KH_2_PO_4_, 0.1% Tween, pH 7.4) and analyzed for FITC–dextran concentration with a fluorescence spectrophotometer at the excitation wavelength of 495 nm and the emission wavelength of 525 nm. Standard curves for determining the concentration in the samples were obtained by diluting FITC–dextran in PBS-T.

### 2.4. Macroscopical Evaluation

The level of colonic inflammation and damage was evaluated from photographs based on three parameters: presence of diarrhea, visible fecal blood, and colon wall damage (edema and/or ulceration). The evaluation was performed independently by three researchers blinded to the treatments. Based on the severity, scores from 0–3 were given for each parameter, and the final colitis score was calculated as their average as described previously [16].

### 2.5. Plasma β-Hydroxybutyrate

As a marker of ketosis, the level of plasma BHB was determined with a commercial enzymatic kit (Cayman Chemicals, Ann Arbor, MI, USA).

### 2.6. Histological Analyses

Tissue sections from the distalmost part of the colon were fixed in 4% paraformaldehyde (Thermo Fischer Scientific, Waltham, MA, USA) solution for 36 h and then stored at 4 °C in 70% ethanol until histological processing. The fixed samples were cut into two halves resulting in two longitudinal pieces that were embedded in paraffin and sectioned at 4 μm thickness. The slides were stained with hematoxylin and eosin (HE) dye and evaluated for histological changes. The slides were read blinded to the treatments.

The HE-stained slides were evaluated for the severity of tissue damage and inflammation, which were separately scored. The scoring mainly followed the system described in [15], which employs separate integer scores from 0 to 3 for tissue damage and inflammation summed to a combined score ranging from 0 to 6. In this study, the scoring was modified to account for the generally mild histological alterations in DSS-KD groups. The mildest DSS-induced changes consisting of minimal or mild mucosal damage received a tissue damage score of 0.5 and a mild *lamina propria* mononuclear infiltrate a score of 0.5. Correspondingly, mucosal tissue damage that did not include marked surface epithelial erosions or ulceration received a damage score of 1.5 and non-extensive damage extending beyond mucosa a score of 2.5. Inflammatory cell infiltration limited to the inner circular layer of *muscularis externa* was scored as 2.5.

### 2.7. Reverse Transcription Quantitative Polymerase Chain Reaction 

The expression of the TJ protein-coding genes *Cldn1*, *Cldn2*, *Cldn4*, and *Ocln* and inflammatory marker genes *Tnf*, *Il1b*, *Il6*, and *Lcn2* in colonic tissue was analyzed with reverse transcription quantitative polymerase chain reaction (RT-qPCR) according to a previously described protocol [14]. In brief, RNA was extracted from colonic samples with NucleoSpin RNA Kit (Macherey Nagel, Duren, Germany). RNA concentration was determined, samples were diluted to the same concentration, and RNA was reverse transcribed to complementary DNA with iScript^TM^ cDNA Synthesis Kit (Bio-Rad, Hercules, CA, USA). RT-qPCR was run with LightCycler^®^ 480 SYBR Green Master (Roche Diagnostics Corp., Indianapolis, IN, USA) with the following amplification protocol: 10 min at 95 °C, 40 cycles of denaturation (15 s, 95 °C), annealing (30 s, 60 °C), and elongation (30 s, 72 °C). The melt curves were analyzed at the end of the experiment. The primer sequences used are listed in Table 2. The results were calculated as relative quantities (RQs) of messenger RNA (mRNA) according to the Vandesompele method [17]. The housekeeping genes used for normalization were *18S*, *Eef2*, and *Rplp0*.

### 2.8. Western Blot

Western Blot (WB) was used to analyze the relative quantity of jejunal and colonic TJ proteins claudin-1, -2, and -4, as well as occludin. Samples were prepared according to a previously described protocol [14]. Briefly, the tissue pieces were homogenized, sonicated, and centrifuged. Supernatants were collected and their protein concentrations determined with a commercial kit (Pierce^TM^ BCA Protein Assay Kit, Thermo Fischer Scientific). Samples for analysis were prepared by diluting supernatants to the same total protein concentration with PBS-T and Laemmli sample buffer (Bio-Rad) containing 5% 2-mercaptoethanol followed by the denaturation of the proteins on a heat block.

From each group, 2–3 samples (30 μg total protein) were loaded in 4–20% Mini-PROTEAN^®^ TGX^TM^ Precast gels (Bio-Rad). After the SDS-PAGE run, the proteins were transferred to a nitrocellulose membrane (Bio-Rad), and total protein concentration was determined with a commercial kit (Revert^TM^ 700 Total Protein Stain Kit, LI-COR, Lincoln, NE, USA). Afterwards, the membranes were blocked (Odyssey blocking buffer (TBS), LI-COR) for 1 h and incubated overnight in a primary antibody solution. The primary antibodies used were claudin-1 (sc-166338, 1:200; Santa Cruz Biotechnology, Dallas, TX, USA), claudin-2 (sc-293233, 1:200; Santa Cruz Biotechnology), claudin-4 (sc-376643, 1:200; Santa Cruz Biotechnology), and occludin (#91131, 1:1000, Cell Signaling Technology, Danvers, MA, USA). The next day, the membranes were incubated in fluorescence-labeled secondary antibody solution (1:10,000 (IRDye 680LT goat anti-mouse, or IRDye 800CW goat anti-rabbit, LI-COR)) for 1 h. The bands were detected with the Odyssey CLx Infrared Imaging system (LI-COR) and analyzed with the Image Studio program (LI-COR). The intensities of the target protein bands were normalized against the intensity of the total protein.

### 2.9. Statistical Analyses

GrapPad Prism 10 (Dotmatics, La Jolla, CA, USA) was used for conducting the statistical analyses and creating the figures. Weight change and DAI were analyzed with mixed-effects model followed by Tukey’s multiple comparisons test. For other parameters, Shapiro–Wilk test was used to determine normal distribution, based on which the data were analyzed either using one-way ANOVA followed by Tukey’s post hoc test or Kruskal–Wallis H test followed by Dunn’s post hoc test. RT-qPCR results are expressed as geometric mean and other data as mean unless stated otherwise. The level of statistical significance was set at *p* < 0.05.

## 3. Results

### 3.1. Ketogenic Diets Mitigate DSS-Induced Weight Loss and Macroscopic Signs of Inflammation

After the induction of colitis, both KDs alleviated inflammation-induced weight loss when compared to DSS-CD in which the mice had lost 20% of their pre-DSS weight at the end of the experiment (Figure 2A). The weight loss in the DSS-SFA-KD group (10% of the pre-DSS weight) was significant when compared to healthy controls which did not lose weight, whereas the DSS-LA-KD group only lost 5% of its pre-DSS weight and the difference to healthy mice did not reach significance. However, there was no significant difference between the KD groups. Weight development in KD groups was comparable to controls before DSS administration, and there were no significant differences in average daily energy consumption (13.0 ± 0.8 kcal/d in CD and DSS-CD, 14.4 ± 1.2 kcal/d in DSS-SFA-KD, and 15.3 ± 1.4 kcal/d in DSS-LA-KD). Compared to DSS-CD, mice in both KD groups exhibited fewer visible signs of colitis, indicated by the lower DAI scores with significant differences between DSS-CD and DSS-LA-KD from day five of DSS administration onwards and between DSS-CD and DSS-SFA-KD on day four (Figure 2B). DSS-CD mice began to display signs of colitis on day four (difference in DAI compared to CD), and this was delayed in DSS-SFA-KD and DSS-LA-KD groups by one and two days, respectively. As expected, on the day of sacrifice, all mice in KD groups showed significantly elevated plasma BHB levels (Figure 2C) which were significantly higher in DSS-LA-KD when compared to DSS-SFA-KD. Some animals in DSS-CD also showed increased levels of BHB. While macroscopic changes in the colon were clearly visible in DSS-CD and DSS-SFA-KD groups, LA-KD-fed mice were protected from these changes (Figure 2D,E). In DSS-LA-KD, DSS-induced colon shortening was prevented (Figure 2D,F). DSS-SFA-KD mice showed a decrease of 13% in the mean colon length, whereas in DSS-CD group, colons were 21% shorter than in healthy controls.

### 3.2. Ketogenic Diets Protect from DSS-Induced Mucosal Damage

Both KDs substantially alleviated DSS-induced histological lesions and reduced the depth of the tissue damage and inflammation (Figure 3A), with a statistically significant effect in the LA-KD-fed mice (Figure 3B–D). The DSS-CD mice exhibited, in general, marked tissue damage and inflammatory cell infiltration, whereas the DSS-SFA-KD mice showed mild to moderate and the DSS-LA-KD mice showed minimal or mild changes, generally limited to mucosa and submucosa. In the DSS-CD group, the lesions extended to submucosa and muscular layers in four and throughout the intestinal wall in two samples. Typical findings were abundant crypt loss, fibroplasia, and diffuse to focally extensive erosion of the surface epithelium with degeneration and dysplasia of the remaining crypts and epithelium. Moderate to marked mixed (macrophages, neutrophils, and lymphocytes) inflammatory cell infiltrate was present.

The mildly affected six DSS-SFA-KD mice showed minimal to mild degenerative changes in the surface epithelium and variable degeneration and dysplasia of the crypt epithelium, as well as goblet cell hypertrophy and/or slight crypt elongation, dilatation, and increased number of mitoses. In the moderately affected four mice, variable crypt loss and few epithelial erosions were present, and surface epithelial cells showed marked degeneration and dysplasia. The *lamina propria* contained minimal to moderate mononuclear cells (macrophages and lymphocytes) or mixed inflammatory infiltrates, and the mixed inflammation extended to submucosa in two mice and to *muscularis externa* in two mice. In the DSS-LA-KD mice, the minimal or mild histopathological changes were limited to mucosa. The surface epithelium exhibited focal to diffuse and minimal to mild degenerative changes, and the crypt epithelium exhibited minimal to mild degeneration and individual cell loss as well as mild crypt dilatation and/or mild elongation and goblet cell hyperplasia. Minimal to mild mononuclear cell inflammatory infiltrate was present in the *lamina propria*. In one mouse, basophilic, low crypt epithelium, and increased number of goblet cells suggested minimal crypt hyperplasia.

### 3.3. Ketogenic Diet High in Linoleic Acid Prevents Increases in Inflammatory Markers

The transcription of all analyzed inflammatory markers was significantly increased in DSS-CD. LA-KD protected from DSS-induced increases in the mRNA expression of *Il1b* and *Tnf*, and there was no statistical difference in any of the inflammatory markers between DSS-LA-KD and CD (Figure 4). While there were no significant differences between DSS-SFA-KD and CD in *Il6* and *Tnf*, DSS-SFA-KD exhibited a modest increase in *Il1b* and *Lcn2*. There were no statistically significant differences between the two DSS-KD groups.

### 3.4. Ketogenic Diets Normalize DSS-Induced Abnormalities in Tight Junction Protein Expression

DSS increased permeability to FITC–dextran (4 kDa) in all groups and its level did not differ significantly between the DSS groups (Figure 5). In DSS-CD, the expression of claudin-1 and claudin-4 was increased both on mRNA and protein level compared to CD and DSS-LA-KD, respectively (Figure 5). Interestingly, *Cldn2* expression was depressed in DSS-CD, but LA-KD preserved its expression. DSS-associated decrease in occludin levels was prevented by LA-KD, whereas in SFA-KD, its expression was modestly lower.

## 4. Discussion

We investigated the impact of KDs with two different fat sources on DSS-induced experimental colitis. The KDs were high in either SFAs or polyunsaturated LA. We analyzed histological changes, intestinal permeability, TJ protein levels, and inflammatory markers. In our setting, both KDs were protective against inflammation, and the effect was more pronounced with LA-KD. To our knowledge, this is the first time KDs with different fat sources have been compared in the context of experimental colitis. Histological damage and inflammatory markers, *Il6*, *Il1b*, *Tnf*, and *Lcn2*, were markedly reduced particularly in LA-KD-fed mice. Colitis-induced derangements in TJ protein and mRNA expression were normalized by LA-KD. Although SFA-KD led to the amelioration of colitis, the effect was lesser than with LA-KD. In addition, LA-KD resulted in higher plasma levels of the ketone body BHB. Here, we show that KDs reduce the severity of DSS colitis, and their preventive potential is dependent on the dietary fatty acid composition.

DSS administration induced weight loss and visible signs of colitis, as well as macroscopic damage and shortening of the colon in animals fed the control diet. Both KDs alleviated inflammation-related weight loss, and the DAI scores were lower in both KDs when compared to DSS-CD. However, only LA-KD was protective against DSS-induced colon shortening and macroscopic changes. In our previous research, BHB levels were not different between healthy mice on SFA-KD or LA-KD [14], whereas in the present study, we saw significantly higher plasma concentrations in DSS-LA-KD than in DSS-SFA-KD. BHB administered either orally [9,25] or rectally [26] has been reported to alleviate DSS-induced colitis, and it is possible that, in our setting, the greater protective effect from LA-KD is partly a result of higher BHB availability. However, in these studies, plasma BHB levels were not reported, and it is unclear whether the magnitude of the difference in our study is large enough to produce a physiologically meaningful effect. Some animals in DSS-CD also showed elevated levels of plasma BHB which is likely due to inadequate food intake and absorption as a response to the DSS-induced inflammation.

Severe DSS-induced histological lesions, damage, and inflammation were observed in DSS-CD. KDs, especially LA-KD, protected the mice from these changes. This is in line with the findings of Kong et al. [10] and Abdelhady et al. [9] who reported less inflammatory cell infiltration in DSS-treated rodents fed KDs with unknown fat sources. On the other hand, Li et al. [11] showed LA-rich KD to promote immune cell infiltration and epithelial damage in DSS colitis. As expected, in DSS-CD, the transcription of *Tnf*, *Il1b*, *Il6*, and *Lcn2* was upregulated. LA-KD prevented the DSS-induced increase in all the inflammatory markers in colon and lowered *Il1b* and *Tnf* compared to DSS-CD. While SFA-KD maintained the levels of *Il6* and *Tnf* with no differences to healthy controls, the diet was unable to fully protect from the elevation in *Il1b* and *Lcn2*. These results contradict those of Li et al. [10] who reported KD to increase colonic mRNA expression of *Il1b*, *Il6*, and *Tnf* upon exposure to DSS. In our prior research, we saw LA-KD to induce slight increase in colonic *Il6* and *Tnf* in healthy animals [14]. However, based on the present study, it seems that these changes do not translate into pronounced inflammation upon a proinflammatory insult.

After the induction of colitis, both KDs preserved claudin expression on a level similar to the healthy controls. Significant changes were observed in DSS-CD and, interestingly, the mRNA and protein expression of the barrier-sealing claudins 1 and 4 was increased, whereas the transcription of pore-forming claudin-2 was decreased. Accordingly, previous research suggests that, even though claudins 1 and 4 are recognized to be barrier-forming TJ proteins [27,28], their protein expression is higher in biopsies from patients with active inflammatory bowel disease, and claudin-1 level is correlated with the severity of inflammation [29]. In addition, claudin-1 overexpression causes increased susceptibility to DSS-induced colitis and inferior recovery from it [30]. Thus, the upregulation of these claudins in DSS-CD is likely a compensatory mechanism. This compensation might be an explanatory factor for why there were no significant differences between DSS groups in intestinal permeability to FITC–dextran despite the overall protective effects of KDs. The protein expression of occludin was depressed in DSS-CD and DSS-SFA-KD, whereas LA-KD preserved a level of expression equal to healthy controls. Previously, we have demonstrated four-week feeding with SFA-KD but not LA-KD to promote the protein expression of claudins 1 and 4 in healthy animals [14], but based on the current study, this does not appear to be meaningful upon DSS-induced inflammation.

The gut microbiota is an important mediator of the DSS-induced inflammation and the effect of KD, whether protective or predisposing, might be related to its composition [10,11]. While the microbiota was not analyzed in this study, it is possible that changes in its composition are responsible for the different outcomes between groups. As a whole, the KD-related gut microbiota seems to decrease the levels of pro-inflammatory Th17 cells in the intestine [31]. These cells contribute to the severity of colitis [32], and while analyzing immune cell subpopulations was outside the scope of this study, KD-mediated reduction of Th17 cells might be one of the mechanisms behind the protective effect. The fat source of the diet also has an impact on the gut microbiota. In mice fed a diet with 42 E% from fat, the fatty acid composition of the diet determines the composition of the microbiota [33], and the possible differences between the two KD groups in our study might explain the greater protective effect from LA-KD. LA supplementation was recently reported to protect from *Porphyromonas gingivalis*-induced aggravation of DSS-induced colitis by decreasing the Th17/Treg cell ratio [34]. The combined effect of KD and LA on T cell subpopulations may be a factor behind the pronounced benefit from LA-KD.

Overall, our results align with the observations of Kong et al. [10] who reported KD to alleviate DSS-induced colitis via microbiota-dependent reduction in colonic group 3 innate lymphoid cells, and Abdelhady et al. [9] who observed benefits associated with NLRP3 inflammasome inhibition, decreased apoptosis, and increased autophagy. Thus, our findings contradict those of Li et al. [11] who reported KD to worsen outcomes in the same model of colitis. Our study challenges the hypothesis of the difference being related to the duration of the diet intervention, since our experiment was closer in length to that of Li et al. (thirty days) than Kong et al. (16 weeks). While the protocol of DSS administration was slightly different in duration and the concentration of the compound, our design was, again, more resembling that of Li et al. in these aspects. Importantly, our findings indicate that the differences observed between studies may not be a result of the fat source of the diet—while the effect was more pronounced in LA-KD, we saw both KDs to offer protection when compared to the non-ketogenic control diet. Moreover, the KD formula used by Li et al. closely resembled LA-KD which we showed to be more protective than SFA-KD. While investigating the cause of these between-study discrepancies warrants further research, this study supports the notion of KDs holding potential in alleviating intestinal inflammation.

## 5. Conclusions

We observed KDs to reduce the severity of DSS colitis, and the preventive effect seems to be affected by dietary fatty acid composition. KDs alleviated inflammation and maintained colonic TJ protein levels, indicating the prevention of colitis development altogether. KD rich in polyunsaturated fatty acid LA was consistently more effective in alleviating colitis than KD high in SFAs. Thus, KDs may have preventive potential against intestinal inflammation, with the level of effect being dependent on the dietary fat source.

## Figures and Tables

**Figure 1 nutrients-16-01348-f001:**
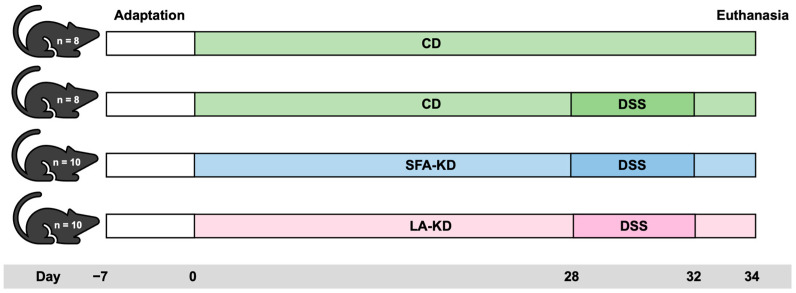
Animal experiment set-up. CD = control diet, DSS = dextran sodium sulfate, LA-KD = ketogenic diet with linoleic acid, and SFA-KD = ketogenic diet with saturated fatty acids.

**Figure 2 nutrients-16-01348-f002:**
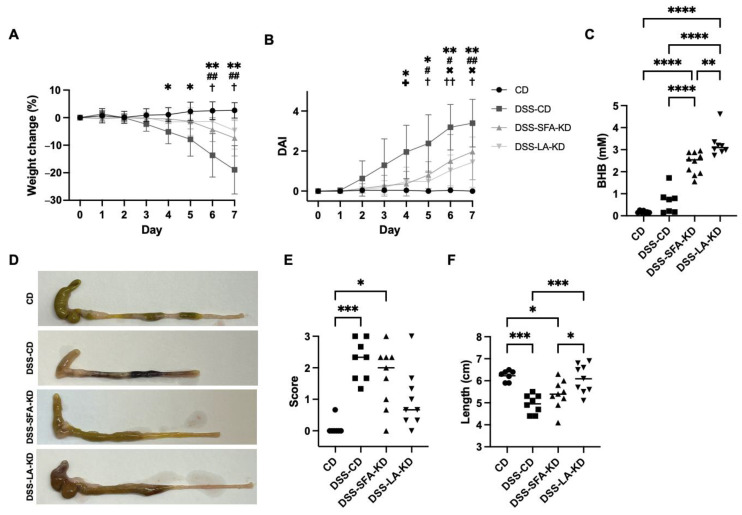
Macroscopic parameters of DSS-induced inflammation and plasma BHB levels. (**A**) Weight change (mean ± SD, *n* = 8–10/group) as a percentage from the start of DSS administration. (**B**) DAI scores (mean ± SD, *n* = 8–10/group) from the start of DSS administration. (**C**) Plasma BHB. (**D**) Representative photographs of colons after DSS period. (**E**) Colitis score from the macroscopic evaluation of the colon. (**F**) Length of the colons after DSS treatment. In (**A**,**B**), * = CD vs. DSS-CD, # = CD vs. DSS-SFA-KD, × = CD vs. DSS-LA-KD, + = DSS-CD vs. DSS-SFA-KD, and † = DSS-CD vs. DSS-LA-KD. * *p* < 0.05, ** *p* < 0.01, *** *p* < 0.001, **** *p* < 0.0001, †† *p* < 0.01, and ## *p* < 0.01. BHB = β-hydroxybutyrate, DAI = disease activity index, DSS = dextran sodium sulfate, CD = healthy group with control diet, DSS-CD = DSS group with control diet, DSS-LA-KD = DSS group with diet high in linoleic acid, and DSS-SFA-KD = DSS group with diet high in saturated fatty acids.

**Figure 3 nutrients-16-01348-f003:**
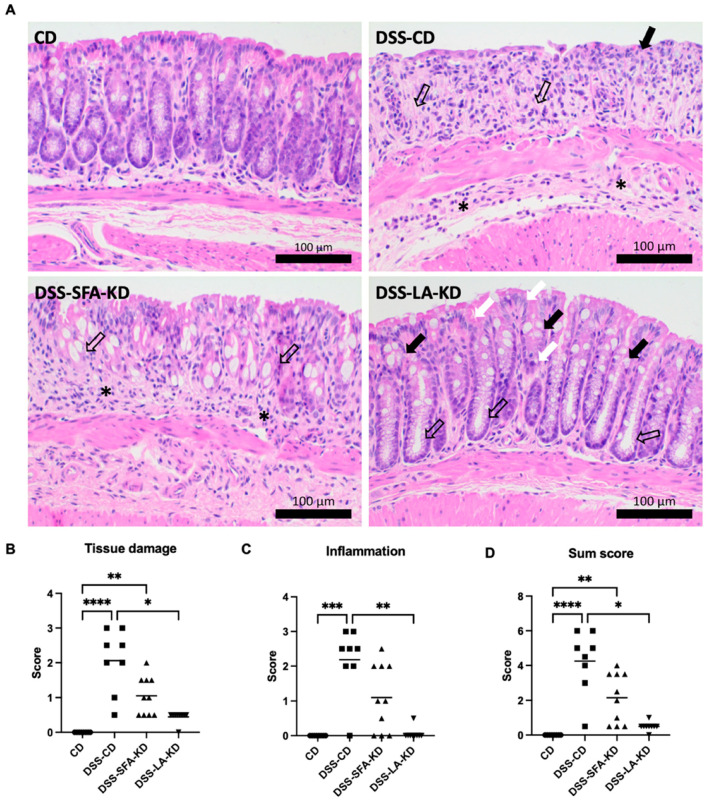
Histological findings and histological lesion scoring of DSS-induced colitis. (**A**) Microphotographs of HE-stained colon sections. Bars 100 µm. CD. No histopathological findings. DSS-CD. Crypt loss and lamina propria fibroplasia (open arrows). Surface epithelium erosion (arrow) and moderate neutrophil and macrophage infiltration in the lamina propria and submucosa (asterisks). DSS-SFA-KD. Marked degeneration and dysplasia of the surface and crypt epithelium and goblet cell hypertrophy (open arrows). Moderate mononuclear cell inflammatory infiltrate in the lamina propria (asterisks). DSS-LA-KD. Minimal degenerative changes in the surface and crypt epithelium (white arrows) and goblet cell hyperplasia (arrows), as well as mild crypt elongation and dilatation (open arrows). (**B**) Tissue damage score. (**C**) Inflammation score. (**D**) Combined sum score of tissue damage and inflammation. Original microphotographs are presented in Appendix A. * *p* < 0.05, ** *p* < 0.01, *** *p* < 0.001, and **** *p* < 0.0001. DSS = dextran sodium sulfate, CD = healthy group with control diet, DSS-CD = DSS group with control diet, DSS-LA-KD = DSS group with diet high in linoleic acid, DSS-SFA-KD = DSS group with diet high in saturated fatty acids, and HE = hematoxylin and eosin.

**Figure 4 nutrients-16-01348-f004:**
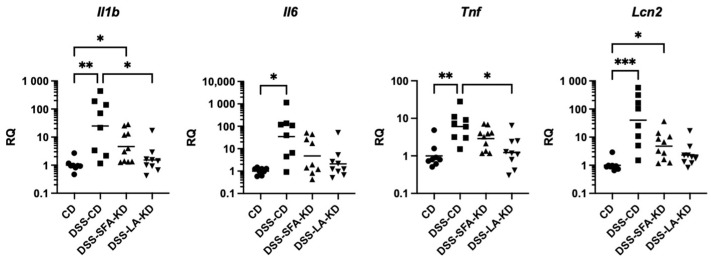
Colonic mRNA expression of inflammatory markers *Il1b*, *Il6*, *Tnf*, and *Lcn2*. * *p* < 0.05, ** *p* < 0.01, and *** *p* < 0.001. DSS = dextran sodium sulfate, CD = healthy group with control diet, DSS-CD = DSS group with control diet, DSS-LA-KD = DSS group with diet high in linoleic acid, DSS-SFA-KD = DSS group with diet high in saturated fatty acids, and RQ = relative quantity.

**Figure 5 nutrients-16-01348-f005:**
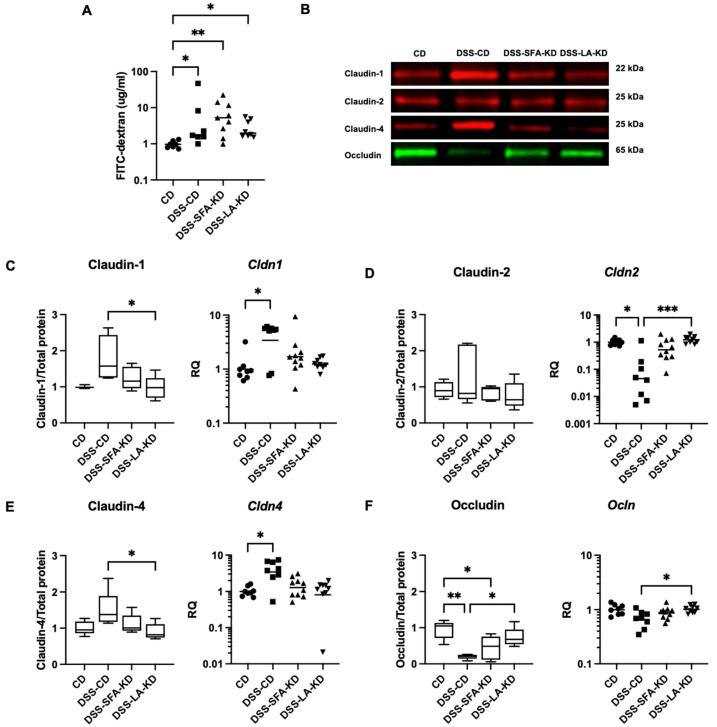
Intestinal permeability to FITC–dextran and colonic tight junction protein expression on protein and mRNA level. (**A**) Intestinal permeability to FITC–dextran (4 kDa) measured as the plasma concentration of the compound. Data is expressed as median. (**B**) Representative Western Blot bands for the analyzed proteins. (**C**) Claudin-1, (**D**) claudin-2, (**E**) claudin-4, and (**F**) occludin. For Western Blot analyses, *n* = 3–5 per group. Images of original blots are presented in Appendix A. * *p* < 0.05, ** *p* < 0.01, and *** *p* < 0.001. DSS = dextran sodium sulfate, CD = healthy group with control diet, DSS-CD = DSS group with control diet, DSS-LA-KD = DSS group with diet high in linoleic acid, DSS-SFA-KD = DSS group with diet high in saturated fatty acids, and FITC–dextran = fluorescein isothiocyanate dextran.

**Table 1 nutrients-16-01348-t001:** The macronutrient compositions of the diets. Values are expressed as a percentage of energy. CD = control diet, SFA-KD = ketogenic diet with saturated fatty acids, and LA-KD = ketogenic diet with linoleic acid.

	CD	SFA-KD	LA-KD
Protein (E%)	9.7	9.4	9.4
Carbohydrate (E%)	77.8	0.5	0.5
Fat (E%)	12.5	90.1	90.1
Saturated fat (E%)	2.6	56.6	22.8
Monounsaturated fat (E%)	3.0	26.0	17.9
Polyunsaturated fat (E%)	6.9	4.5	49.3
Linoleic acid (E%)	6.4	4.0	44.5

**Table 2 nutrients-16-01348-t002:** Primer sequences used in RT-qPCR analyses. If no reference is cited, the primer pair is designed by the authors.

Gene	Forward Primer	Reverse Primer	Ref.
*18S*	AACGAACGAGACTCTGGCAT	ACGCCACTTGTCCCTCTAAG	
*Eef2*	TGTCAGTCATCGCCCATGTG	CATCCTTGCGAGTGTCAGTGA	[18]
*Rplp0*	TAACCCTGAAGTGCTCGACA	GGTACCCGATCTGCAGACA	[19]
*Cldn1*	AGACCTGGATTTGCATCTTGGTG	TGCAACATAGGCAGGACAAGAGTTA	[20]
*Cldn2*	GCAAACAGGCTCCGAAGATACT	GAGATGATGCCCAAGTACAGAG	[21]
*Cldn4*	TGAGCGATGGCGTCTATGG	GATGTTGCTGCCGATGAAGG	
*Ocln*	CGGTACAGCAGCAATGGTAA	CTCCCCACCTGTCGTGTAGT	[22]
*Il1b*	CTCCAGCCAAGCTTCCTTGT	TCATCACTGTCAAAAGGTGGCA	[23]
*Il6*	ATCGTGGAAATGAGAAAAGAGTTGT	CTGCAAGTGCATCATCGTTGT	
*Tnf*	TGGCACCACTAGTTGGTTGTCT	AGCCTGTAGCCCACGTCGTA	[24]
*Lcn2*	CCACCACGGACTACAACCAG	AGCTCCTTGGTTCTTCCATACAG	

## Data Availability

The data used in this study are available from the corresponding author upon reasonable request due to privacy.

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
