# Peer review of "Ketogenic Diet Protects from Experimental Colitis in a Mouse Model Regardless of Dietary Fat Source"

_nutrients, 2024, doi:10.3390/nu16091348_

Round 1

Reviewer 1 Report

Comments and Suggestions for Authors the work is interesting but there are some things to review. I'm not clear on the daily calorie total, please report it clearly. it is not clear to me which protein was used to normalize the wb. why do the authors report only male mice? do they think there are differences with females? I suggest including "colitis" among the keywords. Comments on the Quality of English Language

Minor editing 

Reviewer 2 Report

Comments and Suggestions for Authors

This exciting manuscript studies the effects of fat source on intestinal health in an experimental mouse model of colitis. The comparison groups included low-fat control diet versus saturated fatty acid KD and polyunsaturated linoleic acid KD. Outcome measures included macroscopic and histological changes in the colon, FITC-dextran intestinal permeability, and colon expression of tight junction proteins and inflammatory markers. Both KDs reduced lesions in the colon. Linoleic acid prevented weight loss and shortening of the colon, and mitigated changes in tight junction protein expression. There were no differences in intestinal permeability.

Some issues to address:

Although macronutrient profiles are provided in Table 1, exact diet formulations should be provided in Supplementary Data to facilitate replication of the experiments and comparison with other studies.

Limitations could be added to the Discussion. For example, the tissue was drop fixed and not perfused as blood was also collected. Could the timing of tissue collection and fixation impact histology? The tissue was not flushed prior to sample collection. Could fecal contents impact the western blotting or RTqPCR?

Line 31 check spelling of Chrohn’s disease

Line 206 states weight in KD groups was comparable to controls before DSS-admin This is surprising as other studies find significant weight loss with KD in C57BL/6J mice. Can the authors comment on this? Providing the exact diet formulation in Supplementary Data and comparison with the literature may resolve the difference.
